# High-Efficiency Crystalline Silicon-Based Solar Cells Using Textured TiO_2_ Layer and Plasmonic Nanoparticles

**DOI:** 10.3390/nano12091589

**Published:** 2022-05-07

**Authors:** Ali Elrashidi, Khaled Elleithy

**Affiliations:** 1Department of Electrical Engineering, University of Business and Technology, Jeddah 21432, Saudi Arabia; 2Department of Engineering Physics, Alexandria University, Alexandria 21544, Egypt; 3Department of Computer Science and Engineering, University of Bridgeport, 221 University Ave, Bridgeport, CT 06604, USA; elleithy@bridgeport.edu

**Keywords:** texture TiO_2_, plasmonic nanoparticles, silicon solar cell, short circuit current density, open-circuit voltage, power conversion efficiency

## Abstract

A high-efficiency crystalline silicon-based solar cell in the visible and near-infrared regions is introduced in this paper. A textured TiO_2_ layer grown on top of the active silicon layer and a back reflector with gratings are used to enhance the solar cell performance. The given structure is simulated using the finite difference time domain (FDTD) method to determine the solar cell’s performance. The simulation toolbox calculates the short circuit current density by solving Maxwell’s equation, and the open-circuit voltage will be calculated numerically according to the material parameters. Hence, each simulation process calculates the fill factor and power conversion efficiency numerically. The optimization of the crystalline silicon active layer thickness and the dimensions of the back reflector grating are given in this work. The grating period structure of the Al back reflector is covered with a graphene layer to improve the absorption of the solar cell, where the periodicity, height, and width of the gratings are optimized. Furthermore, the optimum height of the textured TiO_2_ layer is simulated to produce the maximum efficiency using light absorption and short circuit current density. In addition, plasmonic nanoparticles are distributed on the textured surface to enhance the light absorption, with different radii, with radius 50, 75, 100, and 125 nm. The absorbed light energy for different nanoparticle materials, Au, Ag, Al, and Cu, are simulated and compared to determine the best performance. The obtained short circuit current density is 61.9 ma/cm^2^, open-circuit voltage is 0.6 V, fill factor is 0.83, and the power conversion efficiency is 30.6%. The proposed crystalline silicon solar cell improves the short circuit current density by almost 89% and the power conversion efficiency by almost 34%.

## 1. Introduction

The main advantage of crystalline silicon-based solar cells is the stability of the output power compared to other solar cells, the simplicity of the fabrication process, and the availability of the silicon semiconductor material [1,2]. On the other hand, the indirect bandgap of the crystalline silicon reduces the recombination rate, where the recombination process occurs by creating phonons and photons. Hence, the lifetime of the excess carrier is increased, and consequently, the recombination rate is decreased. Furthermore, reducing the recombination rate of the excess carriers leads to reducing the output open-circuit voltage, *V_oc_*, of the solar cell and the solar cell’s overall efficiency, *µ* [3]. Many techniques are used to enhance the light absorbed by the silicon solar cell and consequently the efficiency by using silicon nanowire, SiNWs, or textured surfaces to obtain a multireflection of the incident light [4,5].

The light scattered from texture surfaces is the main reason for high light-harvesting performance [6], where the texturing geometry determines the absorbed wavelength, which is called the resonance wavelength [7]. In addition, graphene, C, the layer, is used to increase the short circuit current density, *J_sc_*, and consequently, the efficiency, which is used for its good electric properties, thin and highly transparent [8,9]. However, using a graphene layer in contact with a silicon layer produces a very low efficiency, which can be improved by increasing the graphene work function by engineering the passivation layer’s interface [10]. Furthermore, a textured titanium dioxide material is used on top of a dye-sensitized solar cell to improve the power absorption and, consequently, the total efficiency [11,12].

Moreover, plasmonic nanoparticles, NPs, are distributed on the solar cell surfaces to enhance the absorption where the absorption depends on the surrounding medium and the NP size, shape, and material [13,14,15]. Furthermore, the surface plasmon localization (SPL) technique of the plasmonic NPs is used as an efficient method to improve the optical absorption of solar cell applications [16,17,18,19,20,21,22,23,24].

In this work, we introduce a silicon-based solar cell using a TiO_2_ texture layer on the top of the structure and graphene layer between the silicon and back reflector layer, and plasmonic NPs distributed on the top of a TiO_2_ surface to enhance the power conversion efficiency (PCE) of the silicon solar cell [10,11,12].

Many related works are introduced to improve the overall efficiency of the silicon solar cell by using SiNWs or texturing the top surface. A summary of those works will be given below.

SiNWs based on a silicon layer have been fabricated by Linwei Yu and Pere Cabarrocas, which improve the overall light absorption [4]. Additionally, and to overcome the light-induced degradation, they introduced a three-dimensional radial junction solar cell. The experimental results show that the maximum *V_oc_* produced from the structure is 0.82 V while the *FF* is 0.73 and *J_sc_* is 15.2 mA/cm^2^ to give a power conversion efficiency of 9.3%.

Swallace et al. produced SiNWs based on a silicon layer covered by an on-site graphene layer for the Schottky junction [8]. The carrier collected was increased by using a graphene layer, which enhances the solar cell efficiency to 3.8%. In this structure, the *V_oc_* is 0.47 V, the *FF* is 0.59, and *J_sc_* is 13.7 mA/cm^2^.

Garnett and Yang produced a large area of SiNWs with radial pn junctions by using a simple fabrication method [25]. The fabricated solar cell was based on a room temperature aqueous etching method and low-temperature thin film deposition with a rapid thermal annealing crystallization process. The overall efficiency is increased to 5.3% at *V_oc_* is 0.56 V, *FF* is 0.607 and *J_sc_* = 17.32 mA/cm^2^.

By transferring printing technique, large-scale silicon solar was fabricated from bulk wafers by Yoon et al. [26]. The feature of the produced device is its mechanical flexibility in addition to the structure transparency and ultrathin microconcentrator designs. The measured open-circuit voltage is 0.51, the fill factor is 0.61, the short circuit current density is 33.6 mA/cm^2^ and the overall efficiency is 11.6%.

Augusto et al. introduced an analytical model for the recombination mechanisms of a silicon solar cell with a low bandgap-voltage offset [27]. The effect of changing excess carrier density on the effective lifetime of the carriers is also given in this work. The untextured structure, with 50 µm thickness, produced a 0.764 V as a *V_oc_*, 0.86 for the *FF*, and 38.8 mA/cm^2^ of *J_sc_*.

Chong et al. used a titanium dioxide (TiO_2_) material on top of silicon thin films for its ability to avoid the reduction in material quality and enhance the surface recombination in the active silicon region [28]. The obtained maximum short circuit current density is 33.4 mA/cm^2^ with a pyramid grating structure for a high absorption in the UV and NIR regions, 400–1000 nm, more than 90% of the incident power.

Texturing the crystalline silicon layer top surface increased the light-harvesting into the silicon solar cell is introduced by Dimitrov and Du [29]. The pyramidal shape texturing enhanced the external quantum efficiency to higher than 90% in the 400–1000 nm with a maximum short circuit current density of 35.5 mA/cm^2^ and overall efficiency of 17.5%.

Zhang et al. presented a high-performance inverted pyramid structure on a mono-crystalline silicon solar cell [30]. Experimental and simulation using the FDTD method are introduced in this work, where the highest obtained efficiency is 22.69% at an open-circuit voltage of 0.68 V, short circuit current density 41.2 mA/cm^2^, and FF is 0.809.

A high-efficiency SiNWs silicon-based solar cell is introduced by Elrashidi [31], which produced a short circuit current density of 41.7 mA/cm^2^ and overall efficiency of 19%. A graphene layer was used to cover the top surface of the NWs, and plasmonic NPs were distributed on top of the silicon-based layer. The obtained open-circuit voltage is 0.63 V, and *FF* is 0.73.

A high-efficiency amorphous hydrogenated silicon solar cell using a nano-crystalline silicon oxide layer is introduced by Khokhar et al. [32]. The given hydride structure shows a significant improvement in the solar cell performance as the open-circuit voltage is 0.724 V, the short circuit current density is 38.95 mA/cm^2^ and the overall efficiency is 21.4%.

This work introduces high-performance silicon-based solar cells using the TiO_2_ texture layer and plasmonic NPs. The short circuit current density, power conversion efficiency, and light absorption are simulated by applying the finite difference time domain method using the Lumerical FDTD solutions software package. The numerical model used in this work is given in the following section, then the effect of changing silicon active layer thickness on the solar cell performance is illustrated. Furthermore, the back reflector grating optimization will be introduced, and then the optimization of TiO_2_ textured layer is also given. Moreover, the performance of the solar cell will be illustrated by using different nanoparticle materials. In addition, a comparison between the proposed solar cell performance and the structures given in the literature is also given in this paper.

## 2. Numerical Analysis

We used the single diode model to calculate the overall solar cell efficiency, PCE. We then applied the Green empirical expression [7], in which the short current density is calculated first using Equation (1). This equation assumes that the incident photon will produce an electron [13].
(1)$$J_{sc}=Q\int I\left(\lambda \right)A\left(\lambda \right)\lambda \;d\lambda \;$$
where *Q* is 0.8 × 10^6^ C·J^−1^·m^−1^, *I*(*λ*) is the spectral irradiance at standard air mass 1.5 (AM1.5), and *A*(*λ*) is the optical absorption at a specific wavelength. Moreover and by considering that the photogenerated excess electron and hole densities are balanced, the values ∆p and ∆n are equal, ∆p=∆n. Furthermore, at a high injection level, we can assume that *n* = *p* = ∆p, and the *V_oc_* of silicon-based solar cell is given by Equation (2) [33].
(2)$$V_{oc}=V_{Th}\;ln\left[{\left(\frac{∆p}{n_{i}}\right)}^{2}\right]\;$$
where $V_{Th}$ is the thermal voltage which is equal to 26 mV, ∆p is the excess carrier concentration for *p*-type ($10^{15}<∆p<2\times 10^{17}{\;\mathrm{cm}}^{-3}$), the given range for injection level of crystalline silicon solar cell [27], and *n_i_* is the intrinsic carrier concentration, $n_{i}=10^{10}{\;\mathrm{cm}}^{-3}$ at room temperature.

Hence, the fill factor can be calculated using Equation (3) [31].
(3)$$FF=\frac{\frac{V_{oc}}{V_{Th}}-\ln\left(\frac{V_{oc}}{V_{Th}}+0.72\right)}{\frac{V_{oc}}{V_{Th}}+1}\;\;$$

The maximum output power, *P_max_*, can be calculated using Equation (4).
(4)$$P_{max}=J_{sc}\times V_{oc}\times FF$$

Hence, the overall solar cell efficiency, *η*, can be calculated as a ratio of maximum output power to solar input power.

Plasmonic nanoparticles distributed on the TiO_2_ textured layer change the absorbed optical power inside the silicon layer, depending on the maximum reflectivity value. NP shape is the main parameter of the transmitted optical power, as well as the relative permittivity of the plasmonic NPs and the dielectric constant of the surrounded medium [31]. The maximum absorption occurred at the maximum value of the wavelength, which can be calculated using Equation (5).
(5)$$\lambda_{max}=\frac{P}{n}{\left(\frac{\varepsilon_{P}\varepsilon_{m}\left(\lambda_{max}\right)}{\varepsilon_{m}+\varepsilon_{P}\left(\lambda_{max}\right)}\right)}^{1/2}$$
where *ε_m_* is the permittivity of the surrounding medium, *ε_P_* is a plasmonic NP dielectric constant at corresponding *λ_max_*, *n* is an integer, and *P* is structural periodicity.

Hence, the dielectric permittivity can be expressed by using a multi-oscillator Drude–Lorentz model [31] as given in Equation (6):(6)$$\varepsilon_{plasmonic}=\varepsilon_{\infty }-\frac{\omega_{D}^{2}}{\omega^{2}+j\omega \gamma_{D}}-\sum_{k=1}^{6}\frac{\delta_{k}\omega_{k}^{2}}{\omega^{2}-\omega_{k}^{2}+2j\omega \gamma_{k}}\;$$
where *ε*_ꝏ_ is the plasmonic high-frequency dielectric permittivity, *ω_D_* and *γ_D_* are the plasma and collision frequencies of the free electrons, *δ_k_* is the amplitude of Lorentz oscillator, *ω_k_* is the angular resonance frequencies, and *γ_k_* is the damping constants for *k* values from 0 to 5.

## 3. Proposed Structure

In this paper, an electromagnetic wave solver, Lumerical finite difference time domain, FDTD, solutions software, is used to design and analyze the introduced structure. A unit cell of the proposed structure was simulated using the FDTD method. The unit cell dimensions are 750 × 750 nm^2^ in the two dimensions, x and y, respectively. A silicon substrate with height is H, which will be optimized lately, where the silicon layer is grown on top of an Al layer as a back reflector, as illustrated in Figure 1. A textured TiO_2_ layer is grown on top of a crystalline silicon layer with height, h1, width, X, and period P, where all will be optimized to obtain the maximum efficiency. Moreover, a periodic grating of Al material will be added to the Al back reflector to improve the light absorption, with period G and height h. In addition to that, a graphene layer can be used to cover the Al grating for more absorption. Finally, plasmonic NPs with different radii will be distributed on the TiO_2_ textured layer, behaving as nano-antennas to retransmit the absorbed light into the TiO_2_ layer.

In order to simulate the proposed structure, the boundary conditions should be defined as well, as we assume a periodic structure in *x*-direction and *y*-direction as the structure is extended in both *x-* and *y*-dimensions. A perfect matching layer in the *z*-direction is considered as the structure is not repeated in the *z*-direction, where the minimum mesh size is 0.25 nm in all directions. A plane wave source with a wavelength band of 400–1100 nm and offset time of 7.5 fs is used as a light source. In addition, the solar generation calculation region is given in the active layer to calculate the short circuit current density.

In this work, we use a simulation method of ideal structure where the effect of local electric fields and the defects at the interfaces on the interfacial recombination, sharpness of grating walls, the position of nanoparticles, and the perfect NPs shape to be able to simulate the proposed structure.

The refractive index of TiO_2_ follows the Devore model [34]; however, the value for silicon is a function of the wavelength. It follows the Aspnes and Studna model [35], and the refractive index of graphene follows the Phillip and Taft model [36]. On the other hand, the refractive index of plasmonic NPs is summarized using Equation (6) in Table 1 [37]. The plasmonic high-frequency dielectric permittivity values are 0.37, 0.86, 0.96, and 0.99 for Ag, Au, Cu, and Al, respectively.

## 4. Results and Discussion

To calculate the open-circuit voltage, Equation (2) is used, considering the value of excess carrier concentration for p-type $10^{15}\;{\mathrm{cm}}^{-3}$, where the *V_oc_* is 0.6 V, and hence *FF*, can be calculated using Equation (3), *FF* is 0.827. First, a simple structure is considered, Al back reflector and silicon active layers, to optimize the silicon layer thickness, H. The silicon thickness is considered from 1–9 µm, and the structure is simulated to obtain light absorption, as shown in Figure 2. As the thickness increases, the absorption increases in the NIR region until H = 8 µm (dark blue curve), then decreases at H = 9 µm (yellow curve).

The obtained results show that increasing active layer thickness leads to an increase in the light absorbed by the solar cell then at high thickness values, H = 9 µm, the absorption is decreased due to the high increase in the active layer volume, which led to recombination of electron–hole pairs before separation.

The short circuit current density and power conversion efficiency for all thickness values are stated in Table 2. The *J_sc_* and PCE increase as H increases, following the absorption curve, then decreases for H = 9 µm. The maximum values are obtained at H = 8 µm for *J_sc_* = 18.8 mA/cm^2^ and PCE = 9.3%.

Grating width, height, and periodicity are very important parameters. They determine the thickness of the active layer and increase the light trapping in the active layer by multireflection of reflected light by the back reflector. Hence, the grating structure in the Al back reflector greatly affects the absorption and, consequently, the power conversion efficiency. So, gratings on the back reflector are designed with height, h, period, G, and width, X, which need to be optimized.

The height of the gratings is changed from 1 µm to 3 µm, with a width from 25 nm to 100 nm at a period range from 150 nm to 250 nm for Si thickness H = 8 µm, as given in Table 3. The maximum performance is obtained at h = 2 µm, X = 50 nm, and G = 200 nm, where the short circuit current density is 37.07 mA/cm^2^ and PCE is 18.3%.

In addition, 3D graphs of the short circuit current density and PCE for grating thickness values, X = 25, 50, 75, and 100 nm, grating period values, G = 150, 200, and 250 nm, and grating period values, h = 1, 2, and 3 µm are illustrated in Figure 3a,b. The *J_sc_* value is obtained at h = 2 µm, X = 50 nm and G = 200 nm as given in Figure 3a, *J_sc_* is 37.07 mA/cm^2^, and PCE is 1.3% obtained from Figure 3b.

Using a graphene layer enhances the absorption and, consequently, the solar cell’s overall performance. The graphene layer has been added on top of the Al back reflector layer, as given in Figure 1. The apposed light when no gratings nor graphene layer is illustrated in Figure 4 compared to absorption in case of gratings without graphene layer and when grating and graphene are used in the structure.

Figure 4 illustrates that the light absorption is improved when the gratings are considered on the Al back reflector and the graphene layer on the top of the Al layer is used, black curve. The short circuit current density is 37.3 mA/cm^2^, and the power conversion efficiency is 18.5%, which is obtained as an effect of using a graphene layer.

Here, the effect of the TiO_2_ textured layer will be studied, as the height of TiO_2_ has a major effect on the solar cell performance. Simulation of a wide range of height is considered, from 0.5 µm to 2.5 µm with step 500 nm as given in Figure 5.

The maximum short circuit current density, 57.03 mA/cm^2^, is obtained at a texture height of 2 µm, and the power conversion efficiency is the maximum value, 28.2%, at the same height. Figure 5 shows that the PCE is increased to the maximum value at 2 µm and then decreases dramatically at 2.5 µm to 18.3%. Hence, the best performance is obtained at a texture height of 2 µm, and the unit cell dimensions are 750 × 750 nm^2^.

Hence, plasmonic nanoparticles are distributed on the TiO_2_ layer to improve the light absorption in UV and NIR regions. Plasmonic NPs are used to absorb the incident light and retransmit it into the active layer, where the size of the nanoparticles determines the peak wavelength absorption. Al NPs are simulated for different radii, 50–125 nm, distributed on the textured surface to obtain the performance of the solar cell.

Figure 6 illustrates that the absorptance for Al NP radius equal to 100 nm gives the maximum absorption black curve. However, NP radius 50, 75, and 125 nm give lower absorption and, consequently, the short circuit current and PCE.

Table 4 shows that the short circuit current density at Al NP radius is 61.9 mA/cm^2^ and the power conversion efficiency is 30.6% at NP radius 100 nm. As the radius of Al NPs increases, the solar cell performance increases up to *R* = 100 nm, then increasing the radius will decrease the performance. The reason for such behavior is that the maximum peak absorption wavelength depends on NP size, surrounding medium as given in Equation (5), and at high radii, the peak wavelength is shifted toward IR, and the absorption will be decreased in the study region.

The proposed structure consists of an Al back reflector layer with gratings of dimensions h = 2 µm, G = 200 nm, and X = 50 nm. A silicon layer with H = 8 µm is placed on the top of Al layer covered by a graphene layer, then a TiO_2_ textured layer with dimensions 0.75 × 0.75 × 2 µm^3^. Finally, Al NPs with radii 100 nm are distributed on the top of the TiO_2_ textured layer to obtain an overall efficiency of 30.6%.

The transmitted, reflected, and absorbed power of the given structure are shown in Figure 7, where the transmitted and reflected power are minimal with respect to the absorbed power.

As shown in Figure 7, the absorbance is more than 90% in around 80% of the wavelength band (0.4–0.955 µm), and more than 80% of the absorbance in 95% of the wavelength band (0.4–1.07 µm), which indicate a high absorption of the proposed structure. The bandwidth at 90% absorption and above is 555 nm, while the bandwidth for 80% absorption and above is 670 nm, which gives a very high bandwidth.

Furthermore, the effect of changing the plasmonic NPs material is also given in this work. Cu, Au, Ag, and Al NPs are simulated to obtain the absorptance then the short circuit current density and PCE are calculated. Figure 8 illustrates the absorptance of the different NPs, which clearly shows that the absorption in the case of using Al NPs, green curve, is higher than the other NPs.

Table 5 shows the short circuit density and power conversion efficiency produced from different plasmonic materials. Al NPs give the higher short circuit current density, 61.9 mA/cm^2^, and consequently, the PCE value is 30.6%. Au, Cu, and Ag give close values for the *J_sc_*, 55.8, 56.2, and 54.6 mA/cm^2^, respectively, where the PCE values are 27.6%, 27.8%, and 27.0%, respectively.

Finally, the proposed silicon-based solar cell is compared to other structures, fabricated structures as in [4,8,25,26,29,32], analytical analysis work as in [27], experimental and simulation analysis as in [30] and simulation analysis as in [28,31], as shown in the literature, as illustrated in Table 6 below. Different structures are given in Table 6, with SiNWs, bulk Si, TiO_2_ textured layer, inverted pyramid Si, and a-Si:H with their output performance. The maximum open-circuit voltage is obtained by Yu et al., 0.82 V, and the maximum FF is 0.86, given by Augusto et al., 0.86. However, the maximum short circuit current density and consequently the power conversion efficiency are produced by our proposed structure with values of 61.9 mA/cm^2^ and 30.6%, respectively. The introduced structure uses the textured TiO_2_ layer for multireflection of the incident light, which is absorbed by the plasmonic nanoparticles and prevents the light from scattering. Hence, plasmonic NPs absorb the incident light and retransmit the light into the active layer of the solar cell, and then the transmitted light is reflected by using the back reflector.

In brief, the main advantage of this work is that the proposed structure enhances the absorption of the silicon-based solar cell in UV and NIR regions, which leads to improving the overall solar cell efficiency.

## 5. Conclusions

This paper introduces plasmonic NPs distributed on a textured TiO_2_ layer grown on top of a crystalline silicon layer to produce a high-performance solar cell. A grating back reflector of Al material is used, where the grating height, 2 µm, grating periodicity, 200 nm, and grating width, 50 nm, are optimized using an FDTD method. The silicon active layer thickness is considered one of the main parameters affecting solar cell performance and needs to be also optimized. The optimum value is calculated to be 8 µm. Textured TiO_2_ with 750 nm length and 750 nm width is grown on the top of the silicon layer, where the texture height is simulated for best performance and equal to 2 µm. Furthermore, plasmonic NPs are distributed on the top of the textured surface with different radii, where the optimum value of NP radius is 100 nm. In addition, different NP materials are also tested, where Al, Ag, Au, and Cu are used as plasmonic NP. Al NPs give the best performance of the solar cell with optical power efficiency = 30.6% for *J_sc_* = 61.9 mA/cm^2^, *V_oc_* = 0.6 V and *FF* = 0.83.

## Figures and Tables

**Figure 1 nanomaterials-12-01589-f001:**
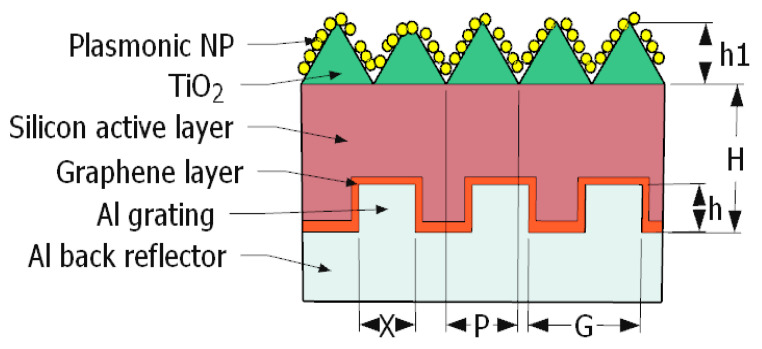
Schematic diagram of the proposed structure.

**Figure 2 nanomaterials-12-01589-f002:**
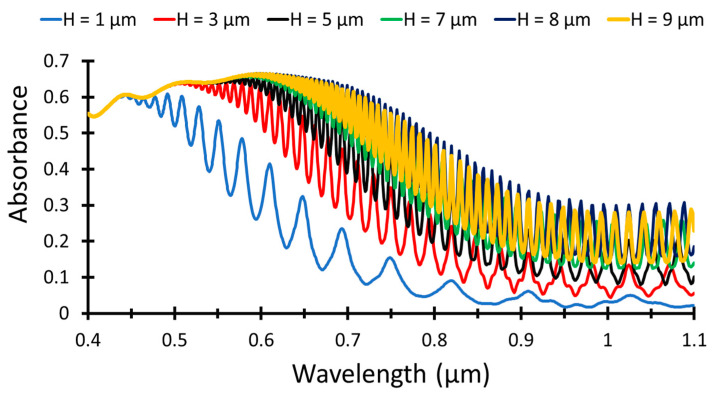
Light absorption for different Si thicknesses as a function of the wavelength.

**Figure 3 nanomaterials-12-01589-f003:**
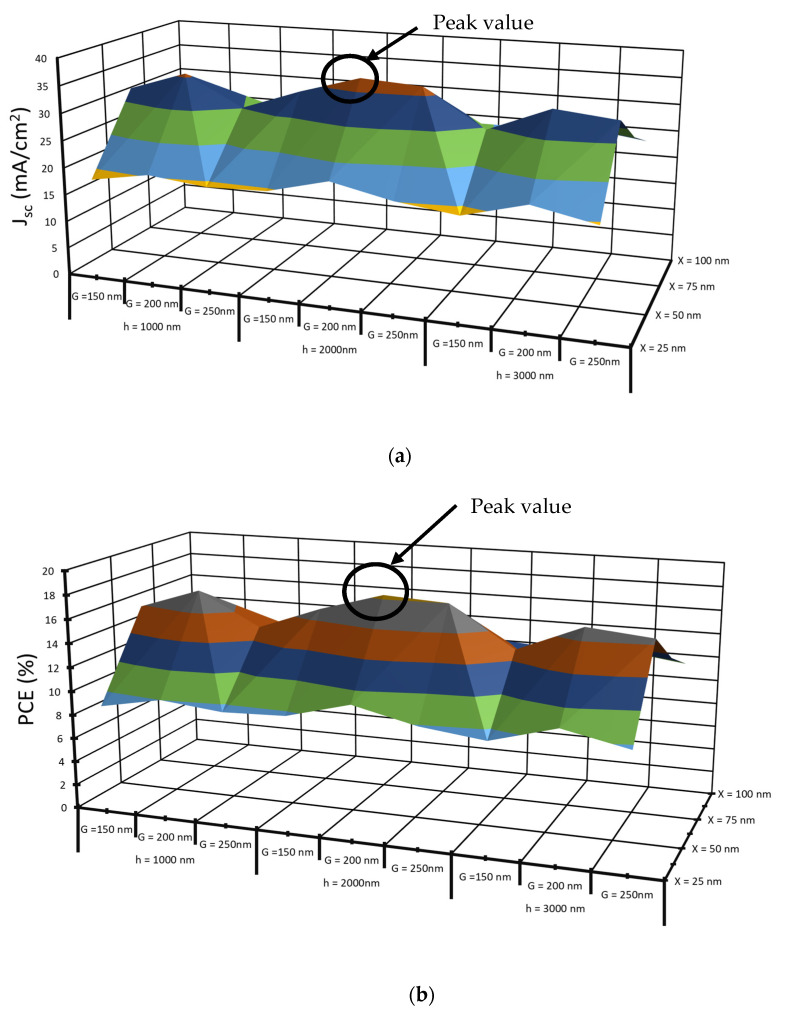
3D illustration of (**a**) short circuit current density and (**b**) PCE as a function of grating width, X, grating height, h, and grating period, G.

**Figure 4 nanomaterials-12-01589-f004:**
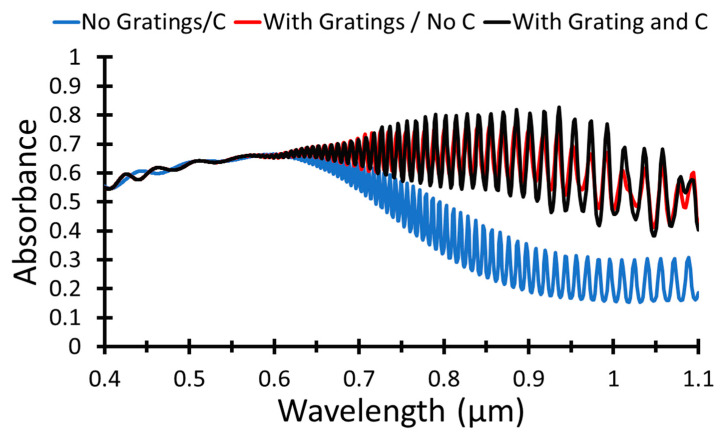
Absorbance when for gratings structure and with/without graphene layer as a function of wavelength.

**Figure 5 nanomaterials-12-01589-f005:**
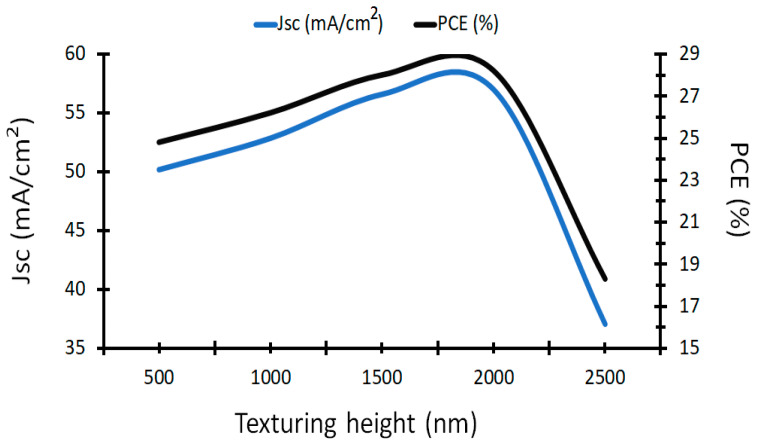
*J_sc_* and PCE for TiO_2_ textured layer (0.5–2.5 µm).

**Figure 6 nanomaterials-12-01589-f006:**
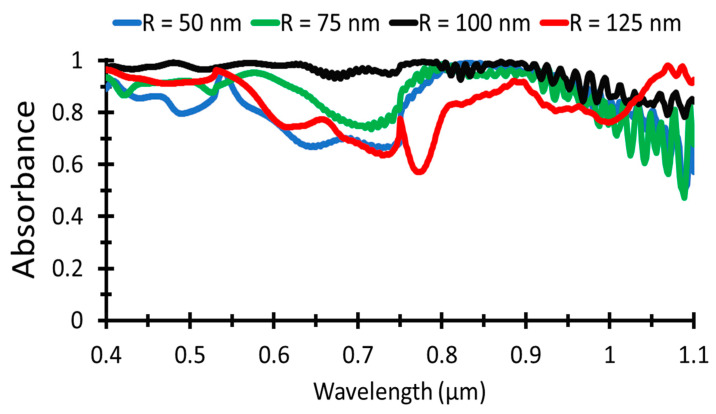
Absorbance for different Al NPs radii as a function of wavelength.

**Figure 7 nanomaterials-12-01589-f007:**
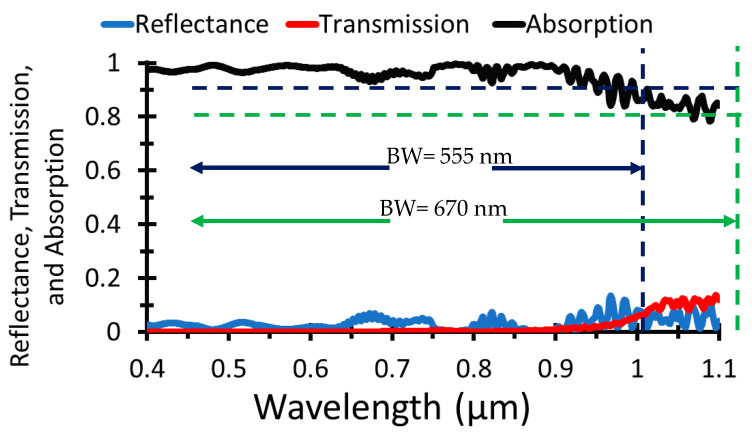
Absorbed, transmitted, and reflected power for the proposed structure.

**Figure 8 nanomaterials-12-01589-f008:**
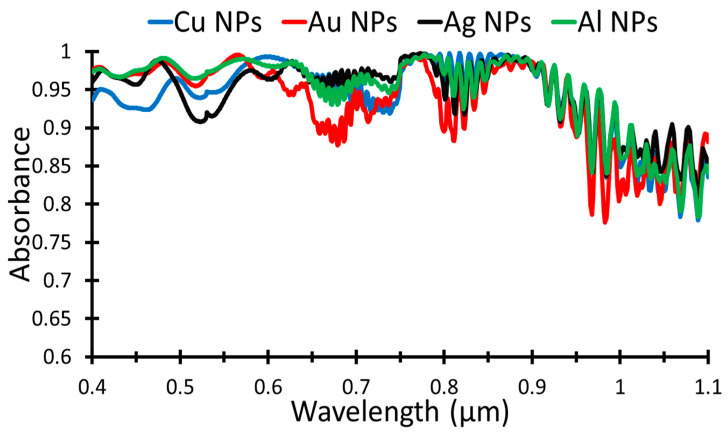
Absorbance for different plasmonic NPS, Cu, Au, Ag, and Al.

**Table 1 nanomaterials-12-01589-t001:** Plasmonic parameters used for the metallic nanoparticle.

Material	Term (k)	Damping Constant	Plasma Frequency (Hz)	Resonant Frequency (Hz)	Damping Frequency (Hz)
Ag	0	0.8450	0.136884 × 10^17^	0.000000 × 10^00^	0.729239 × 10^14^
	1	0.0650	0.136884 × 10^17^	0.123971 × 10^16^	0.590380 × 10^16^
	2	0.1240	0.136884 × 10^17^	0.680775 × 10^16^	0.686701 × 10^15^
	3	0.0110	0.136884 × 10^17^	0.124351 × 10^17^	0.987512 × 10^14^
	4	0.8400	0.136884 × 10^17^	0.137993 × 10^17^	0.139163 × 10^16^
	5	5.6460	0.136884 × 10^17^	0.308256 × 10^17^	0.367506 × 10^16^
Au	0	0.7600	0.137188 × 10^17^	0.000000 × 10^00^	0.805202 × 10^14^
	1	0.0240	0.137188 × 10^17^	0.630488 × 10^15^	0.366139 × 10^15^
	2	0.0100	0.137188 × 10^17^	0.126098 × 10^16^	0.524141 × 10^15^
	3	0.0710	0.137188 × 10^17^	0.451065 × 10^16^	0.132175 × 10^16^
	4	0.6010	0.137188 × 10^17^	0.653885 × 10^16^	0.378901 × 10^16^
	5	4.3840	0.137188 × 10^17^	0.202364 × 10^17^	0.336362 × 10^16^
Cu	0	0.5750	0.164535 × 10^17^	0.000000 × 10^00^	0.455775 × 10^14^
	1	0.0610	0.164535 × 10^17^	0.442101 × 10^15^	0.574276 × 10^15^
	2	0.1040	0.164535 × 10^17^	0.449242 × 10^16^	0.160433 × 10^16^
	3	0.7230	0.164535 × 10^17^	0.805202 × 10^16^	0.488135 × 10^16^
	4	0.6380	0.164535 × 10^17^	0.169852 × 10^17^	0.654037 × 10^16^
Al	0	0.5230	0.227583 × 10^17^	0.000000 × 10^00^	0.714047 × 10^14^
	1	0.2270	0.227583 × 10^17^	0.246118 × 10^15^	0.505910 × 10^15^
	2	0.0500	0.227583 × 10^17^	0.234572 × 10^16^	0.474006 × 10^15^
	3	0.1660	0.227583 × 10^17^	0.274680 × 10^16^	0.205251 × 10^16^
	4	0.0300	0.227583 × 10^17^	0.527635 × 10^16^	0.513810 × 10^16^

**Table 2 nanomaterials-12-01589-t002:** Short circuit current density and PCE for different Si thicknesses.

H (µm)	*J_sc_* (mA/cm^2^)	PCE (%)
1	8.9	4.4
2	11.94	5.9
3	13.6	6.7
4	15.5	7.7
5	16.1	7.9
6	17.1	8.4
7	17.3	8.6
8	18.8	9.3
9	18.1	9.0

**Table 3 nanomaterials-12-01589-t003:** Effect of grating’s height, width, and period on short circuit current density and PCE.

		X = 25 nm		X = 50 nm		X = 75 nm		X = 100 nm	
h (nm)	G (nm)	*J_sc_* (mA/cm^2^)	PCE (%)	*J_sc_* (mA/cm^2^)	PCE (%)	*J_sc_* (mA/cm^2^)	PCE (%)	*J_sc_* (mA/cm^2^)	PCE (%)
1000	150	18.4	9	32.1	15.9	26.2	13	20.9	10.3
	200	20.3	10	35.6	17.6	28.2	14	23.1	11.4
	250	19.2	9.5	30.1	14.9	27.1	13.4	21.6	10.7
2000	150	19.6	9.7	33.9	16.8	26.01	12.9	22	10.9
	200	22.4	11.1	37.07	18.3	28.7	14.2	24.19	12
	250	20.1	9.9	36.4	18	27.1	13.9	23.3	11.5
3000	150	18.8	9.3	29.7	14.7	24.36	12.1	23.4	11.6
	200	21.8	10.8	34.2	16.9	28	13.9	24.1	11.9
	250	19.6	9.7	33.1	16.4	26.8	13.3	22.6	11.2

**Table 4 nanomaterials-12-01589-t004:** Short circuit current density and PCE at different Al NPs radii.

R (nm)	*J_sc_* (mA/cm^2^)	PCE (%)
50	39.8	19.7
75	45.8	22.7
100	61.9	30.6
125	35.5	17.6

**Table 5 nanomaterials-12-01589-t005:** Short circuit current density and PCE for different NP materials.

Plasmonic Material	*J_sc_* (mA/cm^2^)	PCE (%)
Al	61.9	30.6
Au	55.8	27.6
Ag	54.6	27.0
Cu	56.2	27.8

**Table 6 nanomaterials-12-01589-t006:** Comparison between other solar cells and the proposed structure.

Different Structure	Structure	*V_oc_* (V)	*FF*	*J_sc_* (mA/cm^2^)	PCE (%)
[4]	SiNWs	0.82	0.73	15.2	9.30
[8]	SiNWs	0.47	0.59	13.7	3.80
[25]	SiNWs	0.56	0.61	17.3	5.31
[26]	Bulk Si	0.51	0.61	33.6	11.61
[27]	Bulk Si	0.76	0.86	38.8	-
[28]	Bulk Si (textured TiO_2_)	-	-	33.4	-
[29]	Textured Si	-	-	35.5	17.50
[30]	Inverted pyramid Si	0.68	0.81	41.7	22.69
[31]	SiNWs	0.63	0.73	41.7	19.00
[32]	a-Si:H	0.72	-	38.95	21.4
Proposed structure	Texturing Tio_2_ Al NPs Al gratings	0.60	0.83	61.9	30.6

## Data Availability

Not applicable.

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
