# Peer review of "High-Efficiency Crystalline Silicon-Based Solar Cells Using Textured TiO2 Layer and Plasmonic Nanoparticles"

_nanomaterials, 2022, doi:10.3390/nano12091589_

Round 1

Reviewer 1 Report

This manuscript reports that a textured TiO2 layer grown on top of the active silicon layer and a Al back reflector with gratings are used to enhance the solar cell performance. The given structure is simulated using the finite difference time domain (FDTD) method to determine the solar cell's performance. In addition, Au, Ag, Al, and Cu plasmonic nanoparticles are distributed on the textured surface to enhance the light absorption, with different radii, with radius 50, 75, 100, and 125 nm. This study is an interesting work to the readers. However, some other comments are needed to address as follows:     

  1. Page 1/12, Line 35-37: the authors claimed that “On the other hand, the indirect bandgap of the crystalline silicon reduces the recombination rate and open-circuit voltage, Voc, which reduces the solar cell's overall efficiency, µ [3]”. Please give the more information that why the indirect bandgap of the crystalline silicon reduces the recombination rate and open-circuit voltage. Please also check the correct symbol of “µ” for the solar cell's overall efficiency.
  2. Please check the correctness of Equation (2), particularly to the term (Dp/ni )2.
  3. Please give the unit of Plasma frequency, Resonant frequency, and Damping frequency on the Table 1.
  4. Please give the definition of strength as showing in Table 1, on the revise manuscript.
  5. The authors claimed that “However, the maximum short circuit current density and consequently the power conversion efficiency are produced by our proposed structure with values of 61.9 mA/cm2 and 30.6%, respectively”. Can authors to identify what kinds of contribution (textured TiO2 layer, or back reflector with gratings, or plasmonic nanoparticles, or all combinations) on the short circuit current density or power conversion efficiency to achieves such high values?

Reviewer 2 Report

Elrashidi et al., proposed a device architecture for Silicon solar cells by combining Textured TiO2 layer, plasmic metal particles, grating type aluminum back reflectors to improve absorption of the Silicon solar cells from UV- NIR region regions and thereby increase the device performances. The authors used FDTD method to simulate the device characteristics and thereby optimized the device parameters.

This article will be interesting for the readers of the OPV community. Hence this work maybe accepted for publication after minor revisions.

The manuscript is not technically sound. The author should provide more interpretation of their results. It will be important to understand the root causes of the performances and failures of each parameters. Authors should shed light on that.

In my impression authors performed optimization of parameters in a sequential approach and obtained a set of localized parameters. Is it possible to obtain global parameters for each of the components by simultaneously optimizing for the parameters of all the components in their proposed structure. 

The simulations conducted in this work are too ideal. The effects of local electric fields and defects at the interfaces on the interfacial recombination were completely ignored.

Round 2

Reviewer 1 Report

This manuscript reports that a textured TiO2 layer grown on top of the active silicon layer and an Al back reflector with gratings are used to enhance the solar cell performance. This study is an interesting work to the readers. The revised manuscript has been reached:

  1. The authors have addressed my concerns and the manuscript was revised
  2. The quality of revised manuscript has been improved.